# Superficial Vein Thrombosis in an Asymptomatic Case of Cholangiocarcinoma with Recent History of COVID-19

**DOI:** 10.3390/life14091095

**Published:** 2024-08-30

**Authors:** Mihai-Lucian Ciobica, Bianca-Andreea Sandulescu, Mihai Alexandru Sotcan, Lucian-Marius-Florin Dumitrescu, Lucian-George Eftimie, Cezar-Ionut Calin, Mihaela Iordache, Dragos Cuzino, Mara Carsote, Claudiu Nistor, Ana-Maria Radu

**Affiliations:** 1Department of Internal Medicine and Gastroenterology, “Carol Davila” University of Medicine and Pharmacy, 020021 Bucharest, Romania; lucian.ciobica@umfcd.ro (M.-L.C.); bianca-andreea.sandulescu@drd.umfcd.ro (B.-A.S.); ana-maria.radu@rez.umfcd.ro (A.-M.R.); 2Department of Internal Medicine I and Rheumatology, “Dr. Carol Davila” Central Military University Emergency Hospital, 010825 Bucharest, Romania; 3PhD Doctoral School of “Carol Davila” University of Medicine and Pharmacy, 020021 Bucharest, Romania; 4Department of Haematology, “Dr. Carol Davila” Central Military University Emergency Hospital, 010825 Bucharest, Romania; sotcanm@yahoo.com; 5Department of Pathology, “Dr. Carol Davila” Central Military University Emergency Hospital, 010825 Bucharest, Romania; marius_med@yahoo.com (L.-M.-F.D.); lucian.eftimie@unefs.ro (L.-G.E.); 6Discipline of Anatomy and Biomechanics, National University of Physical Education and Sports, 060057 Bucharest, Romania; 7Department of Oncology, “Dr. Carol Davila” Central Military University Emergency Hospital, 010825 Bucharest, Romania; caesar.calin@gmail.com; 81st Internal Medicine Department, “Dr. Carol Davila” Central Military University Emergency Hospital, 010825 Bucharest, Romania; mihaelaiordache2005@yahoo.com; 9Laboratory of Radiology and Medical Imaging II, “Dr. Carol Davila” Central Military University Emergency Hospital, 010825 Bucharest, Romania; dragos.cuzino@umfcd.ro; 10Department of Radiology, Oncology, and Haematology, “Carol Davila” University of Medicine and Pharmacy, 020021 Bucharest, Romania; 11Department of Endocrinology, “Carol Davila” University of Medicine and Pharmacy, 050474 Bucharest, Romania; 12Department of Clinical Endocrinology V, C.I. Parhon National Institute of Endocrinology, 011863 Bucharest, Romania; 13Department 4—Cardio-Thoracic Pathology, Thoracic Surgery II Discipline, “Carol Davila” University of Medicine and Pharmacy, 050474 Bucharest, Romania; 14Thoracic Surgery Department, “Dr. Carol Davila” Central Emergency University Military Hospital, 010825 Bucharest, Romania

**Keywords:** cholangiocarcinoma, liver, thrombosis, paraneoplastic syndrome, surgery, COVID-19, tumour, cancer, biopsy, endocrine surgery

## Abstract

The COVID-19 pandemic brought into prominence several emergent medical and surgical entities, but, also, it served as trigger and contributor for numerous apparently unrelated ailments such as arterial and venous thromboembolic complications. Additional risk factors for these thrombotic traits may be concurrent (known or unknown) malignancies, including at hepatic level. Among these, cholangiocarcinoma (CCA), a rare cancer of intra- and extra-hepatic biliary ducts, represents a very aggressive condition that typically associates local and distant advanced stages on first presentation requiring a prompt diagnosis and a stratified management. This neoplasia has been reported to present a large spectrum of paraneoplastic syndromes in terms of dermatologic, renal, systemic, neurologic, endocrine, and cardiovascular settings, that, overall, are exceptional in their epidemiologic impact when compared to other cancers. Our aim was to introduce a most unusual case of CCA-associated distant thrombosis in a male adult who initially was considered to experience COVID-19-related thrombotic features while having a history of obesity and bariatric surgery. This is a hybrid type of paper: this clinical vignette is accompanied by two distinct sample-focused analyses as a basis for discussion; they each had different methods depending on their current level of statistical evidence. We only included English-published articles in PubMed, as follows: Firstly, we conducted a search of reports similar to the present case, regarding distant vein thrombosis in CCA, from inception until the present time. We performed a literature search using the keywords “cholangiocarcinoma”, “thrombosis”, and “Trousseau’s syndrome” and identified 20 cases across 19 original papers; hence, the current level of evidence remains very low Secondly, we searched for the highest level of statistical evidence concerning the diagnosis of venous thrombosis/thromboembolism in patients who underwent COVID-19 infection (key search terms were “COVID-19”, alternatively, “coronavirus”, and “SARS-CoV-2”, and “thrombosis”, alternatively, “thromboembolism”) and included the most recent systematic reviews and meta-analyses that were published in 2024 (from 1 January 2024 until 8 July 2024). After excluding data on vaccination against coronavirus or long COVID-19 syndrome, we identified six such articles. To conclude, we presented a probably unique case of malignancy with an initial manifestation consisting of recurrent superficial vein thrombosis under anticoagulation therapy, with no gastrointestinal manifestations, in a patient with a notable history for multiple episodes of SARS-CoV-2 infection and a prior endocrine (gastric) surgery. To our knowledge, this is the first identification of a CCA under these specific circumstances.

## 1. Introduction

The COVID-19 (coronavirus disease 2019) pandemic brought into prominence several emergent medical and surgical entities related to the viral infection, a novel aetiology in prior known conditions such as COVID-19-related pneumothorax or subacute thyroiditis, etc., but it also served as trigger and contributor for numerous apparently unrelated ailments such as arterial and venous thromboembolic complications and their clinical manifestations like stroke, myocardial infarction, and deep vein thrombosis, superficial thrombophlebitis, and pulmonary embolism [1,2,3].

Coagulation anomalies in COVID-19 (that clinically embraced a heterogeneous pattern in terms of traditional thromboembolic events, microvascular thrombosis, or disseminated intravascular coagulation, etc.) involve a complex and dynamic pathogenic background underlying a multitude of factors such as, for instance, endothelial dysfunction, hepatic disturbances, pro-inflammatory status, damage to the immune response (including the presence of the cytokine storm in dramatic cases), platelet-, and fibrinolysis-associated disturbances, etc. [4,5,6]. In addition, a hypothesis that thrombosis is responsible for developing post-COVID-19 syndrome was released and it is currently under evaluation [7].

Moreover, apart from the direct and indirect effects of the virus within the human body, the prior or concurrent physiological and pathological conditions of one person might act as additional risk factors for the thrombotic traits such as advanced age, gender differences via sex hormone configuration, vascular and valve dysfunction, other co-infections, or different prior/synchronous known or unknown malignancies [8,9,10]. Essentially, every patient with a severe form of COVID-19 infection or requiring hospitalisation should be assessed with concern to the thromboembolic risk, but the decision regarding anticoagulation should be carefully taken [11,12,13].

On the other hand, most malignant neoplasias, including at hepatic level, are recognized to have an elevated risk of either local thromboses or distant (paraneoplastic) manifestations such as deep vein thrombosis or superficial thrombophlebitis. Among these, cholangiocarcinoma (CCA), a rare cancer of intra- and extra-hepatic biliary ducts, represents a very aggressive condition that typically associates local and distant advanced stages on first presentation requiring a prompt diagnosis and a stratified management, including a liver transplant in selected cases [14,15,16]. The tumour has been reported to present a large spectrum of paraneoplastic syndromes in terms of dermatologic, renal, systemic, neurologic, endocrine, and cardiovascular settings, that, overall, are exceptional in their epidemiologic impact when compared to other malignancies that are generally more often found such as pulmonary or mammary, etc. [17,18,19].

Our aim was to introduce a most unusual case of CCA-associated distant thrombosis in a male adult who initially was considered to experience COVID-19-related thrombotic features while having a history of obesity and bariatric surgery.

This is a hybrid type of paper: this clinical vignette is accompanied by two distinct sample-focused analyses as a basis for discussion; they each had different methods depending on their current level of statistical evidence. We only included English-published articles in PubMed, as follows: Firstly, we conducted a search of reports similar with the present case, regarding distant vein thrombosis in CCA, from inception until present time. We performed a literature search using the keywords “cholangiocarcinoma”, “thrombosis”, and “Trousseau’s syndrome” and identified 20 cases across 19 original papers; hence, the current level of evidence remains very low. Secondly, we searched the highest level of statistical evidence concerning the diagnosis of venous thrombosis/thromboembolism in patients who underwent COVID-19 infection (key search terms were “COVID-19”, alternatively, “coronavirus”, and “SARS-CoV-2”, and “thrombosis”, alternatively, “thromboembolism”) and included the most recent systematic reviews and meta-analyses (n = 6) that were published in 2024 (from 1 January 2024 until 8 July 2024). We excluded data on vaccination against coronavirus or long COVID-19 syndrome.

## 2. Clinical Vignette

### 2.1. Admission

In March 2023, a 44-year-old male was referred by his primary care physician after he underwent a check-up abdominal ultrasound that revealed multiple hypoechoic liver masses; two of them displaced hepatic segments VII and VIII and were considered to be liver metastases. The ultrasound exam was performed in the context of recurrent superficial vein thrombosis of the left upper limb while the patient was under treatment with apixaban, 5 mg twice daily (BID).

### 2.2. Medical History

His records revealed three episodes of COVID-19 infection in the previous two years (of medium severity, without hospitalisation); of note, he was not vaccinated against coronavirus. Soon after the latest episode (September 2022), the subject developed deep vein thrombosis of the lower limbs, affecting the left posterior tibial and soleal veins, the left popliteal vein, and the right posterior tibial veins, and superficial thrombophlebitis involving the right forearm. The thromboses were considered to be related to the recent COVID-19 infection and he was offered apixaban, 10 mg BID, for seven days, followed by 5 mg BID for another six months.

On admission, the subject was under this therapy. Additionally, he was treated with bisoprolol 2.5 mg BID for a recently confirmed sinus tachycardia, as well as naproxen/esomeprazole, chlorzoxazone, and two supplements containing vitamins and antioxidants for a recently diagnosed mild cervical degenerative disc disease at C5–C6 level. Other findings in his medical history were a laparoscopic cholecystectomy for gallstones in 2006, a vertical sleeve gastrectomy for weight loss in 2010, and an osteosynthesis procedure following a traumatic clavicle fracture in 2018.

His family history revealed that his mother had been diagnosed with breast cancer (which was currently in remission) and his father suffered from cardiovascular disease (no further data were available).

The patient worked as a programmer, was a non-smoker, and denied any alcohol consumption.

### 2.3. Physical Examination

At the time of admission, his main complaint was pain in his left forearm, where he had signs of superficial vein thrombosis. He later recounted feeling a little more tired than usual and having trouble sleeping within the last two months.

He was 181 cm tall and weighed 80 kg (body mass index of 24.42 kg/m^2^) after having weighed 130 kg (a body mass index of 39.68 kg/m^2^). He first tried losing weight by having bariatric surgery (a gastric sleeve procedure) in 2010 with limited results. In 2019 he followed a ketogenic diet for six months and lost almost 45 kg, after which he started intermittent fasting (which he was still following) and maintained his weight at around 80 kg.

Clinical exam showed he had slightly pale teguments that were cold in the distal segments of all limbs (without apparent jaundice). As mentioned, his left forearm presented signs of superficial thrombophlebitis, with erythema, induration, and pain along the course of the superficial veins. Indurated cordlike venous segments (without erythema or tenderness) were palpated on both arms and on the right forearm. Heart and lung auscultation was normal with a blood pressure of 115/70 mmHg, a heart rate of 80 bpm, and oxygen saturation of 96% on room air. His abdomen presented loose excess skin and white striae amid his prior weight loss. He denied any bowel transit or urinary complains.

### 2.4. Laboratory Findings

On admission, blood assays identified leucocytosis (white blood cells of 16,590 cells/µL) with increased levels of neutrophils (12,250 cells/µL), monocytes (9500 cells/µL), and eosinophils (1370 cells/µL). Also, a mild hypochromic microcytic hypo-regenerative anaemia (a haemoglobin level of 11.8 g/dL) with a serum iron level of 28 µg/dL (reference range is between 70 and 180 µg/dL) and a ferritin level of 291 µg/L (reference range is between 20 and 250 µg/L) was detected. INR (International Normalized Ratio) value was 1.71; a highly increased D-dimer value of 388 ng/mL (reference range between 0 and 250 ng/dL) was associated with elevated C-reactive protein (CRP) of 31.09 mg/L (reference range between 0 and 5 mg/L), but with normal values of ESR (erythrocyte sedimentation rate) and fibrinogen. A mild cholestasis was identified based on high gamma-glutamyl transferase (GGT) of 65 U/L (reference range between 5 and 55 U/L) and alkaline phosphatase (ALP) of 144 U/L (reference range between 30 and 120 U/L). Lipid profile assays showed hypercholesterolemia in terms of increased total serum cholesterol of 260 mg/dL and LDL (low-density lipoprotein) cholesterol of 189 mg/dL. LDH (lactate dehydrogenase) was also slightly raised, with a value of 329 U/L (reference range between 25 and 248 U/L). Viral hepatitis B and C tests were negative, as were antiphospholipid syndrome tests. Tumour markers carcinoembryonic antigen (CEA), alpha-fetoprotein (AFP), cancer antigen 19-9 (CA19-9), and prostate-specific antigen (PSA) had normal values. Notably, CYFRA21-1 (cytokeratin 19 fragments) had a value of 35.92 ng/mL (reference range is between 0 and 2.08 ng/mL).

### 2.5. Imagery Scans

Venous ultrasound of the limbs described old venous thrombosis, with a hypoechoic material inside the left radial and ulnar veins and recanalization between the thrombus and vein wall. The thrombus also extended towards the brachial vein, with partial thrombosis up to the middle part of the left arm. An affluent of the brachial vein was partially thrombosed on its course between the fist and the elbow. The cephalic and basilic veins also had newly formed thrombotic material in their lumen extending from the level of the forearm up to the distal third of the arm. The right upper limb had no deep vein thrombosis. A superficial vein of the right forearm was dilated and presented minimal post-thrombotic changes. Post-thrombotic changes were described in the veins of the lower limbs as well, involving the veins mentioned in the patient’s medical history.

Trans-abdominal ultrasound identified a liver of increased size, with multiple heterogeneous nodules in both lobes, of round-oval shape, having relatively well-defined margins, with an isoechoic pattern to the surrounding liver parenchyma, and an intensely hypoechoic peripheral halo and size up to 5 cm (aspects that were highly suggestive for liver metastases). In the right liver lobe (segments VII and VIII), a predominantly hypoechoic, poorly defined mass of 8.5 by 5.5 cm was confirmed, with peripheral colour Doppler signal. The mass included a sub-capsular nodular heterogeneous image of approximately 2 cm at the level of segment VI, adjacent to the right kidney, with a hypoechoic ring and discrete peripheral colour Doppler signal (Figure 1).

A contrast-enhanced ultrasound was then performed using SonoVue contrast agent (Bracco, Milano, Italy), focusing on the mass identified in the right hepatic lobe. In the arterial phase, there was a poor peripheral enhancement, with no enhancement at the centre of the mass and fast wash-out, leading to an intense hypo-enhancing aspect at the end of the arterial phase that persisted during the portal-venous and late phases. The smaller lesions were assessed as late phase and revealed an intense hypo-enhancement when compared to the surrounding liver parenchyma. These features were suggestive for an intrahepatic CCA with multiple metastases within the same organ (Figure 2).

A contrast-enhanced computed tomography scan of the thorax, abdomen, and pelvis identified multiple bilateral pulmonary nodules measuring up to 0.8 cm, absence of mediastinal lymph node involvement, and a partial filling defect of the right inferior lobar pulmonary artery, extending into the posterior basal segment and into the medial arterial segment of the middle lobe, suggesting chronic thrombosis without parenchymal infarction. A small non-circumferential pericardial effusion was also identified.

At abdominal level, the computed tomography scan identified an enlarged liver (cranio-caudal diameter of 20 cm) with a large mass (within the segments VII to VIII) of 12 by 10 by 6.7 cm. The mass had irregular, polycyclic margins and had an inhomogeneous structure because of central necrosis. After contrast media injection, the tumour showed peripheral enhancement progressing towards the centre (Figure 3).

The tumour compressed the inferior vena cava that had a diameter of 1.1 cm at this level, and seemed to extend into the right adrenal gland, which was enlarged (a tumour-like appearance, of 3 by 1.6 cm); no clear cleavage plan was detected between the cave vein and the mentioned endocrine structure. These traits suggested an atypical CCA, but a hepatic metastasis of unknown origin could not be entirely excluded. There were also multiple smaller tumours with similar characteristics, considered to be secondary tumours; they had a diffuse distribution inside the liver, with the largest one being localized in the caudate lobe that was distorted and enlarged (5.8 by 5.2 cm). There was no dilatation of intrahepatic biliary ducts and no portal vein thrombosis. Lymphadenopathies were identified in the hepatic hilum, portacaval space, and paracaval space. There was also a small accumulation of fluid in the pelvis, with a thickness of 1.9 cm.

### 2.6. Diagnosis Procedures

Upper gastrointestinal endoscopy and colonoscopy were performed to exclude a gastro-intestinal tract neoplasm and associated metastases. The upper endoscopy revealed extension of the gastric mucosa into the oesophagus, a post-gastric sleeve scar on the greater curvature of the stomach, and hyperaemic mucosa. Oesophageal mucosal biopsies were performed. The colonoscopy identified internal haemorrhoids and two sessile polyps of 0.3 to 0.4 cm, which were endoscopically removed. Pathological exam of the oesophageal fragments identified intestinal metaplasia, while the two polyps revealed mixed inflammatory infiltrate with eosinophils in the lamina propria.

A percutaneous liver biopsy was performed and two fragments from the tumour mass at the left hepatic lobe were sent for a pathological exam that highlighted the diagnosis of adenocarcinoma (no other specification). In addition, an immunohistochemistry exam was performed and identified positive CK7 (cytokeratin), CK19, and CA19-9 markers, with a Ki67 proliferation index of 30%. Hence, a final diagnosis of poorly differentiated adenocarcinoma, most probably a CCA, was established (Figure 4).

### 2.7. Management

The imaging studies (ultrasound and computed tomography) identified a hepatic tumour of increased size located in segments VII and VIII and multiple other nodular lesions with similar characteristics, considered to be secondary tumours. According to the TNM classification, the case was evaluated as cT4cN1cM1 (multiple liver metastases and possible secondary lesions in the lungs), meaning stage IV cholangiocarcinoma.

During hospitalisation, treatment with low-molecular-weight heparin was initiated, with slow remission of the superficial thrombophlebitis. Anticoagulant administration was withheld before the gastrointestinal endoscopies and percutaneous liver biopsy, without incidental thrombotic or haemorrhagic events.

According to the CCA diagnosis, the tumour was considered inoperable by the multidisciplinary team and further chemotherapy was initiated, GEMOX (gemcitabine oxaliplatin), with medium tolerance and no other thrombotic events during the next 3 months.

Of note, a contrast-enhanced computed tomography scan of the thorax, abdomen, and pelvis identified multiple bilateral pulmonary nodules measuring up to 8 mm, absence of mediastinal lymphadenopathy, and a partial filling defect of the right inferior lobar pulmonary artery, extending into the posterior basal segment and into the medial arterial segment of the middle lobe, suggesting chronic thrombosis without parenchymal infarction. No biopsy was recommended by the multidisciplinary team and the lung lesions remained stationary during follow-up. The computed tomography scan of the brain did not reveal any lesions.

## 3. Discussion

Similar cases of patients facing venous thromboembolism while being confirmed with a malignancy and recently experienced a COVID-19 infection have been reported [20], but to our knowledge this is the first identification of a CCA under these specific circumstances. Moreover, the medical history revealed other potential contributors to the thrombotic events (long term obesity, unsuccessful bariatric surgery, cholecystectomy for stones) and even an adrenal metastasis (but this entity does not have a pathological confirmation, only a suspicion based on scan imagery-). Other work-ups in hypercoagulability status such as assessment of factor V Leiden or *JAK2* (Janus Kinase 2) pathogenic variants were not analysed [21,22,23,24]. Which exactly is the role of each element is an open matter, including the interpretation of the time frame from coronavirus infection until the confirmation of a malignancy (if unknown), as seen here. Additionally, the decision to use anticoagulants remains challenging in most cases with mixed pro-thrombotic (multidisciplinary) elements [20,25,26,27,28].

Specifically, CCA presented as recurrent superficial vein thrombosis of the left upper limb, with no gastrointestinal manifestations, in a man with a history of vertical sleeve gastrectomy and three episodes of COVID-19 infection that all are essential for the thrombotic events associated with second-line factors, namely, cholelithiasis (that had been surgically treated more than 15 years before the actual CCA diagnosis was made) as well as a prior long history of obesity. Possibly, we may extend this clinical picture to the presence of non-alcoholic fatty liver disease, given its frequent association with obesity and metabolic disorders (no specific hepatic evaluation was performed before the patient lost weight). Of note, the patient was not diabetic, and we mention this aspect since COVID-19 infection was involved in a higher rate of thrombotic events in the diabetic population [28,29,30].

It is also worth noting that, while obesity was associated with multiple types of cancer (e.g., breast, gastro-intestinal tract, pancreatic, hepatic, gallbladder, kidney, uterine, and ovarian), multiple studies confirmed that bariatric surgery (including sleeve gastrectomy) reduced the risk of all cancer types, obesity-related cancer, and cancer-related mortality, and this is directly correlated with the amount of induced weight loss [31,32,33,34,35,36,37,38]. Interestingly, another finding from our patient’s history that might lower the cancer risk was the dietary regimen he followed to lose weight. Emerging evidence showed that intermittent fasting and a ketogenic diet promote metabolic health and reduce cancer risk through multiple mechanisms including anti-oxidative and anti-inflammatory effects, changes in the gut microbiota, and activation of the anti-carcinogenic pathways. Intermittent fasting also may enhance the effects of chemotherapy in different cancer types, while protecting healthy cells from its adverse effects [39,40,41,42].

Of note, although intrahepatic CCA may be asymptomatic in its early stages, the disease was in an advanced stage on admission following the history of COVID-19 infections (no suggestive symptoms such as abdominal pain, jaundice, night sweats, or weight loss were detected) [43,44]. In this case, the superficial venous thromboses developed while the patient was following anticoagulation therapy with apixaban for multiple venous thrombi, diagnosed six months before current presentation as being associated with an episode of SARS-CoV-2 (severe acute respiratory syndrome coronavirus 2) infection. We consider that this first thrombotic event may be an early CCA manifestation and the diagnosis of coronavirus probably delayed the recognition of the malignancy. On the other hand, as mentioned above, thrombotic events, while known to appear in the setting of a visceral malignancy, are less commonly associated with CCA, so in a patient with no other symptoms and a recent episode of SARS-CoV-2 infection, the clinical suspicion of cancer remains very low [18,45,46,47].

### 3.1. Sample-Focused Analysis: Thrombotic Events in CCA

Regarding the association between thrombotic events and CCA, according to our mentioned methods, we identified 20 other cases across 19 original papers [48,49,50,51,52,53,54,55,56,57,58,59,60,61,62,63,64,65,66] (Table 1).

Only six of the twenty cases [48,49,50,51,52,53,54,55,56,57,58,59,60,61,62,63,64,65,66] had a positive outcome, while in most cases, death occurred relatively shortly after hospitalisation. This finding highlights the poor prognosis of this combination of ailments and the importance of early diagnosis and prompt treatment. It is also worth noting that in five cases thrombotic events reoccurred or persisted despite therapeutic anticoagulation with warfarin, as well as with parenteral anticoagulation such as heparin and low-molecular-weight heparin. This suggests that recurrent thrombotic events in the setting of CCA are more likely due to the evolution and characteristics of the tumour rather than the use of anticoagulants, although a significant reduction in venous thromboembolism recurrence in cancer patients treated with this regime compared with conventional treatment with vitamin K antagonists was reported without a significant increase in bleeding-related complications [67,68].

Alternatively, direct oral anticoagulants, namely, edoxaban and rivaroxaban, are useful, except for gastrointestinal cancer [69,70,71]. A recent meta-analysis of four studies concluded that apixaban is also a good option for the treatment of cancer-associated venous thromboembolism; a lower risk of thrombosis recurrence was detected in the apixaban group compared to the low-molecular-weight heparin group, with similar effects regarding the risk of major bleeding or mortality [72]. Contrary to the findings of the previously mentioned meta-analysis, our patient developed recurrent superficial thrombophlebitis under apixaban which resolved under treatment with low-molecular-weight heparin, thus, suggesting that anticoagulant medication should be chosen and adjusted on a case-by-case basis. This case also highlights the importance of follow-up in patients with peripheral venous thrombosis, even in instances presenting an unapparent predisposing element or in the presence of transient risk factors such as COVID-19.

### 3.2. Sample-Focused Analysis: Thrombotic Events and COVID-19

Concerning the association between venous thrombotic events and COVID-19, according to our methods, we identified six systematic reviews/meta-analyses published in 2024, two years following the actual pandemic years [5,9,10,73,74,75] (Table 2).

Overall, an interesting, yet not homogenous panel reported a higher risk of venous thromboembolism in COVID-19 infection, particularly in individuals who displayed severe forms that required hospitalisation [5,9,10,73,74,75]. Current guidelines recommend anticoagulation therapy for three months in cases of venous thromboembolism associated with a transient risk factor (such as an infection), as it was in this case. The recent history of SARS-CoV-2 infection and the absence of additional clinical signs and symptoms suggestive of an associated pathology qualified the initial management of the case as being adequate. Current recommendations advise against extensive evaluation for an occult malignancy, beyond age-appropriate screening, even in the absence of evident triggers [76,77,78].

Data have shown that occult cancer, often in advanced or metastatic stage, may be identified in approximately 5% of unprovoked venous thromboembolism cases [78]. One study reported that 17% of patients with recurrent forms were diagnosed with cancer within two years of follow-up, as compared with only 4.5% of patients with no recurrence of these thrombotic events [79]. Despite these findings, multiple trials investigating the use of computed tomography, endoscopy, ultrasound, and additional laboratory testing for cancer screening in patients with unprovoked thrombotic events could not demonstrate a clear benefit in mortality or morbidity rates, even if a malignancy was confirmed. Current recommendations state that in the absence of other clinically relevant signs or symptoms, cancer screening in patients with a first episode of unprovoked venous thromboembolism should be restricted to a thorough medical history and physical examination, basic laboratory evaluation (e.g., complete blood count, calcium, urinalysis, and liver function tests), and chest X-ray in addition to recommended age- and gender-specific cancer screening (cervix, prostate, lung, and colon) [76,77,78,79]. Even in individuals with recurrent venous thromboembolism under anticoagulation therapy, it is recommended to firstly assess treatment adherence, correct dosing, and possible drug interactions before further evaluation for a malignancy or alternative causes of recurrence [76,79,80,81,82,83].

In this case, we also stress the fact that the first diagnostic tool used in the evaluation of the recurrent superficial venous thrombosis was an abdominal ultrasound, which immediately raised the suspicion of a hepatic malignancy. This method is widely available at any level of primary health care, with a relatively low cost and no exposure to ionizing radiation, and it stands for the best imagery strategy in screening of liver tumours, and even long-term surveillance in certain subgroups [84,85,86].

Notably, a growing body of literature reports a possible link between SARS-CoV-2 infection, severe immune system alterations, and risk of cancer development that should be taken into consideration in this case as similarly described in other oncologic areas. Among possible common pathogenic pathways between the viral infection and a malignancy, we mention those related to extra spindle pole bodies like spindle pole body 1, Holliday junction recognition protein, topoisomerase II alpha, cyclin B2, etc. Further evidence is mandatory. This is in fact a dual interplay since cancer-related depletion of B cells and T cells increased the risk of infection or severe forms of COVID-19 [87,88,89].

## 4. Conclusions

Amid the multi-layered panel of risk factors for venous thrombosis, for instance, CCA confirmation and a recent history of COVID-19 infection, we mention the importance of a complicated index of suspicion; thus, awareness is mandatory. Recognition of CCA presents multiple pitfalls and challenges in daily practice, hence, leading to a positive diagnosis in advanced stages, because of limited clinical signs and symptoms in the absence of obvious bile duct compression elements. On the other hand, this type of tumour may associate a variety of paraneoplastic syndromes that are potentially useful if they are recognized early.

To conclude, we presented a probably unique case of CCA with initial manifestation consisting of recurrent superficial vein thrombosis under anticoagulation therapy, with no gastrointestinal manifestations, in a patient with a notable history for multiple episodes of SARS-CoV-2 infection and a prior endocrine (gastric) surgery.

Additionally, we performed two connected sample-focused analyses that highlighted and placed this interesting vignette amid the literature data and current level of evidence. The case confirms once again the importance of patients’ follow-up with regard to the presence of peripheral venous thrombosis, even in cases associated with an apparent transient risk factor; it points to the utility of abdominal ultrasound in the evaluation of recurrent thrombotic events with no apparent cause and to the role of a multidisciplinary team in the diagnosis of liver masses of unknown aetiology. Similar cases of patients facing venous thromboembolism while being confirmed with a malignancy and having recently experienced a COVID-19 infection have been reported, but, to our knowledge, this is the first identification of a CCA under these specific circumstances.

## Figures and Tables

**Figure 1 life-14-01095-f001:**
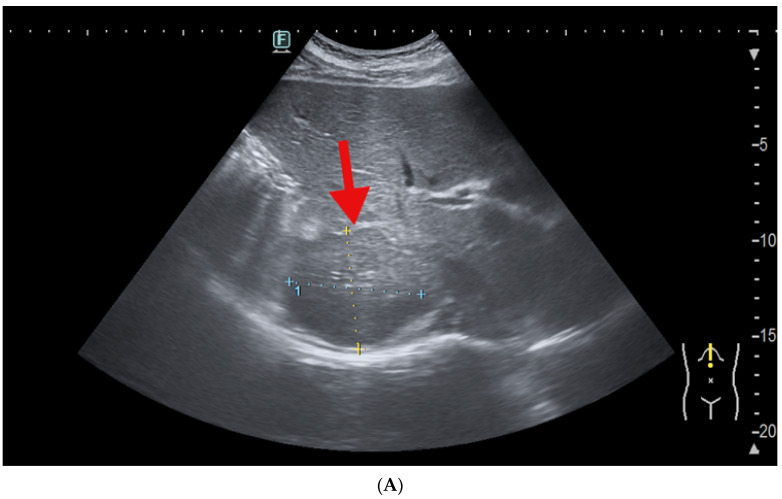
Trans-abdominal ultrasound of the liver in a 44-year-old male with 6-month history of deep vein thrombosis and superficial thrombophlebitis. (**A**) Poorly defined, predominantly hypoechoic tumour mass (red arrow) in the right hepatic lobe (of 8.5 by 5.5 cm). (**B**) Isoechoic liver nodule (intensely hypoechoic peripheral halo)—red arrow.

**Figure 2 life-14-01095-f002:**
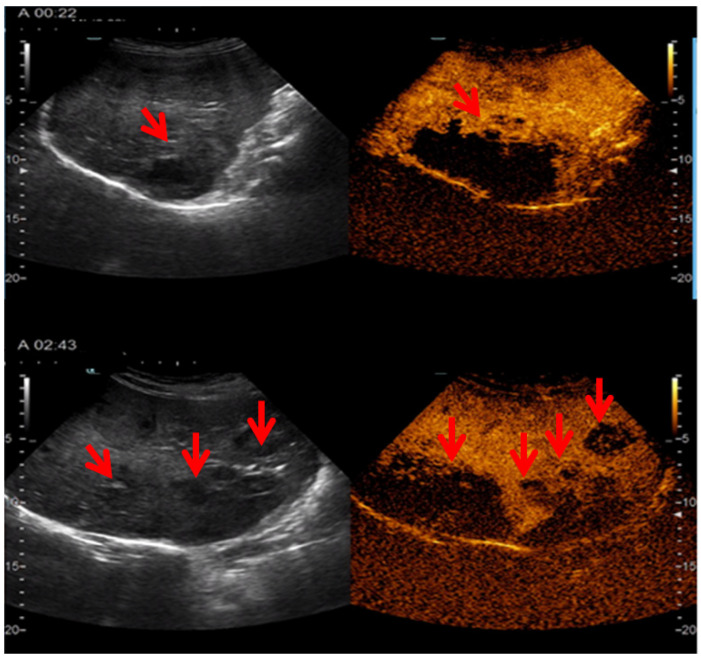
Contrast-enhanced ultrasound (SonoVue contrast agent) aspects: early arterial phase of the tumour mass (red arrow) situated at the level of right liver lobe (**upper capture**) and late arterial phase (**lower capture**) of the smaller disseminated lesions (red arrows); these captures are suggestive for an intrahepatic CCA and multiple metastases at liver level (red arrows show different capture timing).

**Figure 3 life-14-01095-f003:**
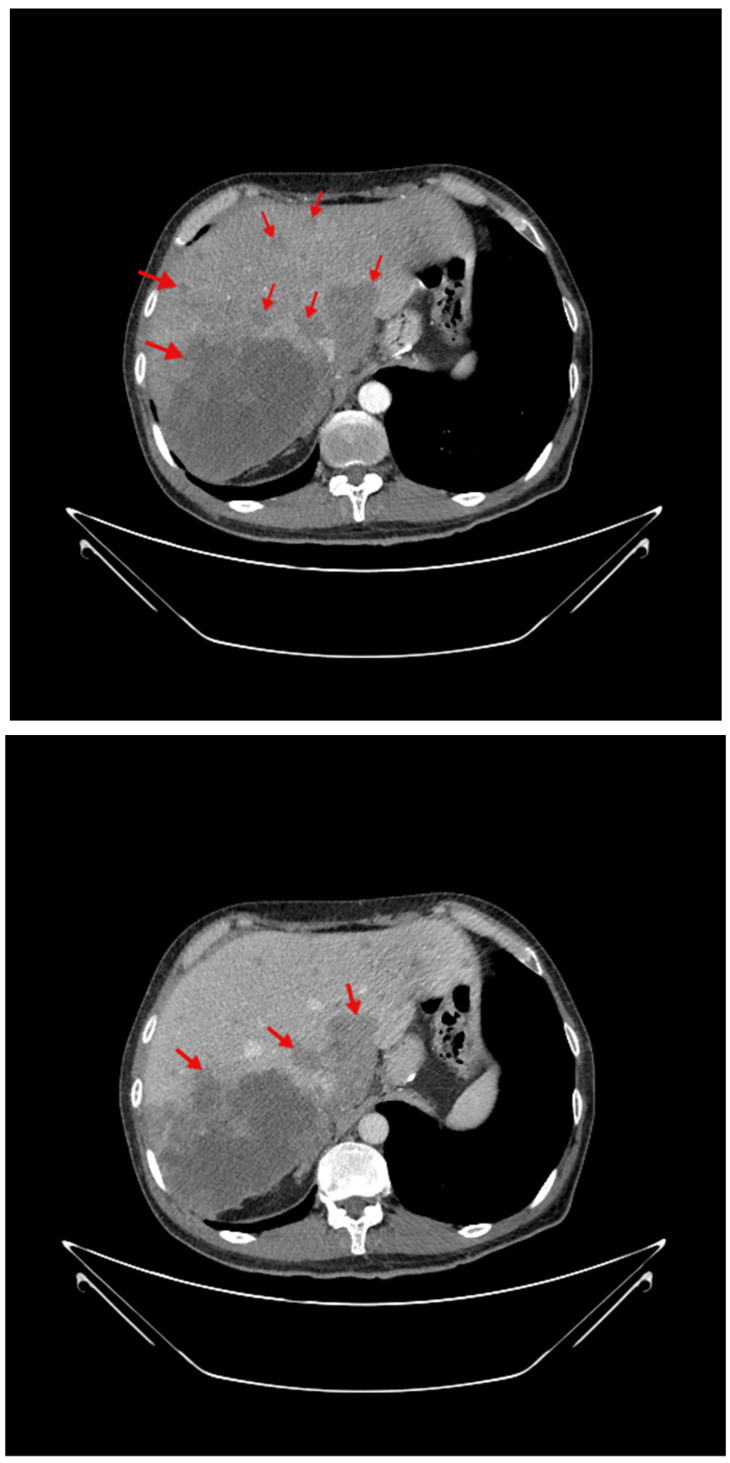
Abdominal contrast-enhanced computed tomography at liver level showing a tumour mass (red arrow) of 12 cm largest diameter; arterial phase (**upper capture**) and venous phase (**lower capture**).

**Figure 4 life-14-01095-f004:**
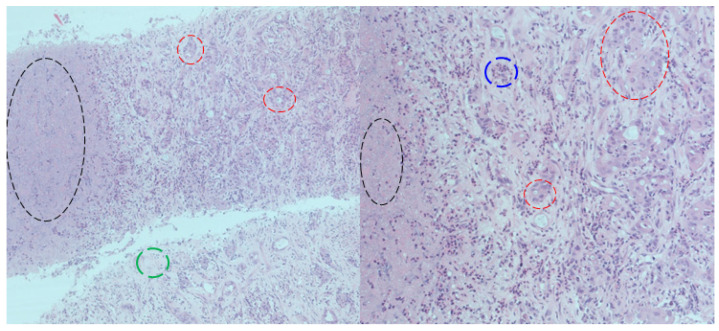
Histological exam via percutaneous liver biopsy of the tumour mass: haematoxylin–eosin staining; absence of the normal liver architecture; areas of necrosis (black highlights), poorly differentiated granular structures with pleomorphic nuclei and altered cell polarity that are highly suggestive for an adenocarcinoma (red highlights); intense desmoplastic reaction (green highlights); mixed inflammatory cells having a diffuse distribution across the entire fragment (blue highlights); (**left capture**—×100 magnification, respectively, **right capture**—×200 magnification).

**Table 1 life-14-01095-t001:** Case reports of thrombotic events associated with cholangiocarcinoma across PubMed search of English language papers; the display starts with the oldest report [48,49,50,51,52,53,54,55,56,57,58,59,60,61,62,63,64,65,66].

First Author Year of Publication Reference Number	Patient’ Age and Sex	Clinical Presentation	Thrombosis Site	Anticoagulation	Outcome
Ching (1991) [48]	76, M	NA	Portal vein	N/A	Death, 5 days later
Martins (1994)—case 1 [49]	67, M	Clinical signs of deep vein thrombosis of right leg (history of primary sclerosing cholangitis, inflammatory bowel disease, and chronic pancreatitis)	Deep vein thrombosis of right leg; 1 month later: superficial thrombophlebitis and deep vein thrombosis of left leg (under warfarin)	Warfarin	Death, 8 weeks later
Martins (1994)—case 2 (1994) [49]	39, M	Clinical signs of deep vein thrombosis of the leg (history of ulcerative colitis and primary sclerosing cholangitis)	Deep vein thrombosis of the leg; migratory superficial thrombophlebitis shortly after	NA	Death, 6 weeks later
Hernandez (1998) [50]	45, F	Sudden dyspnoea	Deep vein thrombosis of left femoral vein; bilateral pulmonary embolism; portal vein and iliac vein thrombosis	Heparin	Death, a few days later
Samadian (1999) [51]	78, F	Pain and swelling in left leg	Deep vein thrombosis of the legs (third episode in 2 years); pulmonary embolism	Warfarin (at presentation INR of 2.2; pulmonary embolism developed while INR was 3)	Death, 2 years after the first thrombotic event
Koskinas (2000) [52]	30, F	Six days of progressive shortness of breath; severe respiratory distress with circulatory collapse on admission (HBsAg positive)	Massive pulmonary embolism	NA	Death, a few hours after admission. Autopsy–combined hepatocellular-cholangiocarcinoma infiltrating the entire liver, metastatic invasion of lung blood vessels
Bandyopadhyay (2003) [53]	38, F	Pain and swelling of left lower limb; 3 months later: two episodes of hematemesis over 1 week and melena; 4 months later: progressive jaundice with pruritus, dull ache in the right upper abdomen	Deep vein thrombosis of left femoral vein; 3 months later, splenic vein thrombosis; 4 months later, multiple thrombi in the portal vein, splenic vein, and intrahepatic portion of inferior vena cava	Warfarin	Patient refused therapy. Of note: normal abdominal ultrasound during the first and second thrombotic episodes
Tasi (2004) [54]	57, F	Right hemiplegia; leg tenderness; 1 month later: recurrent left-side limb weakness and disturbance of consciousness (recent history of CCA treated with surgery and palliative radiotherapy)	Left middle cerebral arterial infarction; deep vein thrombosis of the lower limbs; 1 month later: extensive hyper-acute infarction of right posterior frontotemporal lobe; thrombosis of inferior vena cava and left ventricle	Warfarin–partial resolution of symptoms; 1 month later: enoxaparin–thrombotic process continued despite treatment	Death, 1 month later
Jang (2006) [55]	56, M	Weight loss and mild shortness of breath	Pulmonary embolism	Low-molecular- weight heparin	No thrombosis at 3 months follow-up
Vysetti (2009) [56]	55, M	Worsening left lower extremity pain and swelling of 2 days	Extensive thrombosis in all segments of the deep veins	Heparin	Resolution of symptoms after salvage left iliac thrombectomy
Muñoz-Ortego (2011) [57]	51, F	Fever and decreasing consciousness in the previous 24 h, aphasia, leftward gaze deviation	Multiple strokes; infarct of right lung, left kidney, and spleen; mitral valve vegetation	Heparin (blood tests were positive for lupus anticoagulant and antinuclear antibodies; patient was considered to have catastrophic antiphospholipid syndrome)	Death, unknown timing; on autopsy—lung adenocarcinoma and intrahepatic cholangiocarcinoma
Gnanapandithan (2014) [58]	62, M	Progressively worsening diffuse abdominal pain over 4 days (history of gall bladder carcinoma, right nephrectomy for renal cell carcinoma; anticoagulation with warfarin for mechanical aortic valve)	Extensive portal and superior mesenteric vein thrombosis, which extended under therapy with warfarin	Warfarin (chronic treatment), low-molecular-weight heparin	Discharged on low-molecular-weight heparin and total parenteral nutrition. Of note: a mass in the region of the hepatic hilum was identified only 4 months after the initial presentation, when the patient had developed jaundice
Chang (2014) [59]	47, M	Pain and swelling in left lower extremity (5 days after a 16 h train ride), right sided chest pain, shortness of breath, and haemoptysis (history of self-injections with testosterone)	Proximal common femoral vein; bilateral inferior sub-segmental pulmonary embolism. During hospitalisation also developed deep vein thrombosis of both lower limbs, extending into inferior vena cava; right atrial thrombosis causing acute embolic stroke to the territory of the left middle cerebral artery (through patent foramen oval)	Heparin (patient also received an inferior vena cava filter and regional thrombolysis)	Death, shortly after the stroke
Yuri (2014) [60]	73, F	Light-headedness and dementia of unknown cause for 6 months	Right cortex cerebral infarction; mitral valve vegetation; thrombotic lesions in rectum and spleen	Anticoagulant therapy	Death, 3 months later
Vakil (2015) [61]	64, M	Four-week history of right upper quadrant abdominal pain, early satiety	Extensive portal, splenic superior and inferior mesenteric vein thrombosis with wedge-shaped areas of liver perfusion abnormalities	Warfarin and intravenous heparin	Death, 10 days later. Autopsy revealed extensive, diffuse intrahepatic CCA that had almost replaced normal liver parenchyma
Blum (2016) [62]	69, M	Ten days of worsening right upper quadrant abdominal pain, anorexia, dark urine	Portal vein, pulmonary saddle embolus	Heparin	Death, 1 week later
Dunn (2017) [63]	71, M	Right-sided visual disturbance, vague history of left hand discoloration and parenthesis (history of hypertension and diabetes)	Multiple infarcts in occipital lobe, thalamus and both cerebellar hemispheres; thrombosis of left subclavian artery, segmental pulmonary arteries, and superior mesenteric artery. Of note: no evidence of venous or intra-cardiac thrombosis or thrombophilia	Therapeutic anticoagulation	Good response to chemotherapy and radiation
Zhang (2019) [64]	59, F	Sudden dysfunction of left upper limb with pain and paralysis 8 h before admission; severe dyspnoea after exercise 6 days prior	Left brachial artery; left muscular calf vein; pulmonary embolism: inferior left lobe, right middle, and right lower lobes	Argatroban associated with catheter directed thrombolysis of the brachial artery; Rivaroxaban for systemic anticoagulation	Alive at 6 months, no recurrence of paradoxical embolism or major bleeding (patient was diagnosed with patent foramen oval with atrial septal aneurysm and complete right to left shunt)
Murahashi (2020) [65]	59, F	Upper abdominal pain, nausea, dizziness	Shower embolization under cortex of left temporal lobe and parietal lobe; renal infarction. On day 9: occlusion of proximal right middle cerebral artery	Heparin	Good evolution after right hepatectomy and partial diaphragmatic resection
Sasaki (2020) [66]	59, F	Abdominal pain, headache, and nausea	Splenic and bilateral renal infarction; multiple acute cerebral infarctions; 1 week later: thrombosis of the right middle cerebral artery (under-dosed heparin)	Heparin; low-molecular-weight heparin switch before discharge	Alive at 6 months, with no thrombotic events (following surgical treatment and chemotherapy)

Abbreviations: CCA = cholangiocarcinoma; F = female; INR = international normalized ratio; M = male; NA = not available.

**Table 2 life-14-01095-t002:** Venous thrombotic events and COVID-19 infection: systematic reviews published on PubMed in 2024 (from 1 January until 8 July) [5,9,10,73,74,75].

First Author Reference Number	Study Design	Studied Population	Outcome
Huang [9]	Chart review (1-year single hospital experience in 2020)	N1 = 164 COVID-19 patients; N2 = 492 non-COVID-19 patients (all patients diagnosed with sepsis, were older than 18 years, and were admitted to intensive care unit)	N1 versus N2 were older (*p* = 0.021). N1 versus N2 had higher body mass index (*p* < 0.001). N1 = N2 had similar risk of thrombosis (OR = 0.85; 95%CI: 0.42–1.72). N1 versus N2 had lower risk of mortality (OR = 0.33; 95%CI:0.16–0.66).
Othman [10]	Systematic review	n = 33 studies in hospitalized patients with COVID-19	COVID-19 patients: rate of venous thromboembolism (deep vein thrombosis: 0.4–84%, and pulmonary embolism: 1–40%) was higher versus arterial thromboembolism (stroke: 0.5–15.2%, and myocardial infarction: 0.8–8.7%). All-cause mortality due to thromboembolism complications: 5–48%.
Khoshnegah [5]	Systematic review and meta-analysis	n = 28 studies that provided protein C and S assays in COVID-19 patients	Protein C activity was lower in COVID-19 patients versus controls (pooled *p* = 0.004). Protein S activity was lower in COVID-19 patients versus controls (pooled *p* = 0.002). Protein C activity was lower in non-surviving versus surviving COVID-19 patients (pooled *p* < 0.0001). No association between protein C and S activity and thrombosis risk (*p* > 0.05).
Iam-Arunthai [73]	Retrospective cohort (2-year multicentre experience, between 2021 and 2022)	N = 160 hospitalised patients with COVID-19	Rate of thrombotic complications: 12.5%. Rate of mortality: 36.3%. Low-molecular-weight heparin did not decrease the incidence of venous thromboembolism (risk group stratification).
Algarni [74]	Systematic review of published cases	N = 212 patients with cerebral venous sinus thrombosis and COVID-19	Higher risk for the condition in smokers (mostly men) and women under oral contraceptives or diagnosed with autoimmune diseases. In-hospital mortality rate: 21.3% (overall mortality rate for males: 65.2% versus females 34.8%, *p* = 0.027).
Liu [75]	Systematic review and meta-analysis	n = 17 (from January 2019 to October 2020) N = 7035 COVID-19 patients (weighted mean age of 60.01 years; 62.64% males)	Weighted mean difference for patients with versus without venous thromboembolism: interleukin-6 = 31.15 (95%CI: 9.82–52.49); ferritin = 257.02 (95%CI: 51.7–462.33); LDH = 41.79 (95%: 19.38–102.96) (similar LDH).

Abbreviations: CI = confidence interval; COVID-19 = coronavirus disease 2019; LDH = lactic dehydrogenase; N = number of patients; n = number of studies; OR = odds ratio.

## Data Availability

Other data of the present case are available upon reasonable request.

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
