# Peer review of "Superficial Vein Thrombosis in an Asymptomatic Case of Cholangiocarcinoma with Recent History of COVID-19"

_life, 2024, doi:10.3390/life14091095_

Round 1
Reviewer 1 Report
Comments and Suggestions for Authors
A too long case report (11 authors, more than 4000 words with many useless data).
I am not convinced that VTE in the presented case is due to neoplasia.
Had the patient a PICC line? (distant VTE).
Row 86: Plates?
Row 187:INR is not useful when the patient is treated by NOAC.
Row 307: Which was the daily dose of LMWH? Was the Apixaban stopped?
Row 455: "mostly unique"? perheaps "probably unique"? "mostly unique" can be interpreted as a kind of oxymoron.
Author Response
Response to Review 1 Comments
Dear Reviewer,
Thank you very much for your time and your effort to review our manuscript.
We are very grateful for providing your valuable feedback on the article.
Here is our response and related amendment that has been made in the manuscript according to your review (marked in yellow colour).
A too long case report (11 authors, more than 4000 words with many useless data).
Thank you very much. We respectfully mention that it has been a massive work in the clinical field and related literature. Moreover, according to the MDPI rules there is no limit with regard to the number of authors or to the length of such paper.
Thank you
I am not convinced that VTE in the presented case is due to neoplasia.
Thank you very much. Indeed, we agree. The potential influence of prior COVID-19 infection cannot be excluded, thus the entire presentation and discussion in this specific matter. Moreover, this makes the case even more challenging; hence, the importance of the practical points in this instance. Thank you.
Had the patient a PICC line? (distant VTE).
No, he did not have a PICC line prior to hospital admission. Thank you
Row 86: Plates?
Thank you very much. We corrected it.
Row 187: INR is not useful when the patient is treated by NOAC.
Thank you very much. We respectfully mention the fact that we have checked coagulation parameters as part of the standard initial evaluation of the patient, not to asses NOAC treatment. Thank you
Row 307: Which was the daily dose of LMWH? Was the Apixaban stopped?
Yes, Apixaban was stopped before LMWH treatment, which we decided to use during the hospitalisation in order to perform the hepatic biopsy, which we considered necessary for the definitive diagnosis after the first ultrasound evaluation. Another reason for the switch of anticoagulant was to monitor the evolution of the thrombotic lesions the patient presented with, as well as to assess if new thrombotic events would occur under a different anticoagulant treatment. Thank you
Row 455: "mostly unique"? perheaps "probably unique"? "mostly unique" can be interpreted as a kind of oxymoron.
Thank you very much. We corrected it.
Thank you very much.

Reviewer 2 Report
Comments and Suggestions for Authors
The manuscript by Mihai-Lucian Ciobica et al titled “Cholangiocarcinoma-related distant thrombosis: unravelling 2 the dynamic interplay with COVID-19 infection” reports a case of vascular thrombosis in a middle-age male patient with history of obesity and COVID-19. This is an interesting topic because cancer is an independent risk factor for thrombosis, and in fact diagnosis of unprovoked thrombosis before the diagnosis of cancer is not rare. The authors must have implemented a lot of efforts in preparing this manuscript and I would like to congratulate them for doing so. However, I believe there are areas for improvement which should be addressed before this manuscript is considered for publication. The following are my comments:
1- I believe the title could be improved because this manuscript does not really study all the aspects of the interplay between COVID-19 and cholangiocarcinoma-related distant thrombosis. This is up to the authors, but I think a title like “Superficial vein thrombosis in the left upper limb in an asymptomatic case of cholangiocarcinoma with recent history of COIVID-19 and obesity: A case report” seems more reasonable and appealing. But again, this is just a suggestion.
2- A mix of American spelling (e. g. hospitalization) and British English (e. g. aetiology) has been used. The authors need to use only one style consistently throughout the manuscript.
3- Line 139 – Please notice that SARS-COV2 is the name of the virus whereas the name of the disease is COVID-19. I believe the name of the disease should be used in this line.
4- Line 179 – What does “urinary accuses” mean? I think this might be a typo.
5- In several cases abbreviations for tests (e. g. CRP, ALP, GGT) have been written before the full name. Normally, we expect to see the full names first, followed by the abbreviation in the first instance and then consistent use of the abbreviation afterwards. Please make sure to apply this to the whole manuscript.
6- For Table 1 and Table 2, it is better to write the abbreviations as a footnote for the table.
7- Table 1 – under the column heading of “Patient’s age and sex” please only write age and sex data. Other information should be moved to other columns such as under “clinical presentation”. You may add “and history” to this column’s heading.
8- Table 2, study by Huang et al, under outcome, it mentions: “N1 versus N2 had higher risk of mortality (OR=0.33; 95%CI:0.16-0.66)”. I am confused because when N1 has a higher risk of N2, you would expect to have an OR bigger than 1.0. Could you clarify? Maybe you need to change the wording.
9- It seems the content under section 3.2. Sample-focused analysis: thrombotic events and COVID-19, are out of the scope of the manuscript because they seem to be related to COVID-19 patients, not those with cholangiocarcinoma with a current or recent COVID-19. We know that COVID-19 is associated with increased risk of thrombotic events and morality. I believe this could be mentioned in a paragraph instead of presenting unnecessary results of the literature review. I suggest the table should be removed.
10- Line 421 and 422 for the sentence “Data have shown that occult cancer, often in advanced or metastatic stage …”, please provide a reference.
11- Please do the same (add reference) for the sentence in lines 422 – 425.
12- Line 447, the phrase “panel of potential contributors”, I believe the words “risk factors” would be a better choice of word.
13- There is a need for English editing (moderate). In the pdf file I am attaching, I have highlighted some areas in need of re-writing the sentences or using different words.

The manuscript needs an English edit.
Author Response
Response to Review 2 Comments
Dear Reviewer,
Thank you very much for your time and your effort to review our manuscript.
We are very grateful for providing your valuable feedback on the article.
Here is our response and related amendment that has been made in the manuscript according to your review (marked in yellow colour).
The manuscript by Mihai-Lucian Ciobica et al titled “Cholangiocarcinoma-related distant thrombosis: unravelling 2 the dynamic interplay with COVID-19 infection” reports a case of vascular thrombosis in a middle-age male patient with history of obesity and COVID-19.
This is an interesting topic because cancer is an independent risk factor for thrombosis, and in fact diagnosis of unprovoked thrombosis before the diagnosis of cancer is not rare.
The authors must have implemented a lot of efforts in preparing this manuscript and I would like to congratulate them for doing so.
Thank you very much. We really appreciate it!
However, I believe there are areas for improvement which should be addressed before this manuscript is considered for publication. The following are my comments:
Thank you very much. We addressed them point by point as follows. Thank you
I believe the title could be improved because this manuscript does not really study all the aspects of the interplay between COVID-19 and cholangiocarcinoma-related distant thrombosis. This is up to the authors, but I think a title like “Superficial vein thrombosis in the left upper limb in an asymptomatic case of cholangiocarcinoma with recent history of COIVID-19 and obesity: A case report” seems more reasonable and appealing. But again, this is just a suggestion.
Thank you very much. We changed the title according to your recommendation. Thank you for your excellent suggestion.
A mix of American spelling (e. g. hospitalization) and British English (e. g. aetiology) has been used. The authors need to use only one style consistently throughout the manuscript.
Thank you very much. We corrected the English language. Thank you
Line 139 – Please notice that SARS-COV2 is the name of the virus whereas the name of the disease is COVID-19. I believe the name of the disease should be used in this line.
Thank you very much. We corrected it.
Line 179 – What does “urinary accuses” mean? I think this might be a typo.
Thank you very much. We corrected it.
In several cases abbreviations for tests (e. g. CRP, ALP, GGT) have been written before the full name. Normally, we expect to see the full names first, followed by the abbreviation in the first instance and then consistent use of the abbreviation afterwards. Please make sure to apply this to the whole manuscript.
Thank you very much. We corrected them.
For Table 1 and Table 2, it is better to write the abbreviations as a footnote for the table.
Thank you very much. We corrected them.
Table 1 – under the column heading of “Patient’s age and sex” please only write age and sex data. Other information should be moved to other columns such as under “clinical presentation”. You may add “and history” to this column’s heading.
Thank you very much. We moved them.
Table 2, study by Huang et al, under outcome, it mentions: “N1 versus N2 had higher risk of mortality (OR=0.33; 95%CI:0.16-0.66)”. I am confused because when N1 has a higher risk of N2, you would expect to have an OR bigger than 1.0. Could you clarify? Maybe you need to change the wording.
Thank you very much. We corrected it (“lower” instead of “higher”). Thank you
It seems the content under section 3.2. Sample-focused analysis: thrombotic events and COVID-19, are out of the scope of the manuscript because they seem to be related to COVID-19 patients, not those with cholangiocarcinoma with a current or recent COVID-19. We know that COVID-19 is associated with increased risk of thrombotic events and morality. I believe this could be mentioned in a paragraph instead of presenting unnecessary results of the literature review. I suggest the table should be removed.
Thank you very much. We moved the table as a supplementary table. We respectfully mention that we decided to pinpoint the aspects concerning COVID-19 and thrombosis since they are essential to truly capture the essence of this case (from our point of view). Thank you
Line 421 and 422 for the sentence “Data have shown that occult cancer, often in advanced or metastatic stage …”, please provide a reference.
Thank you very much. We addded it.
Please do the same (add reference) for the sentence in lines 422 – 425.
Thank you very much. We addded it.
Line 447, the phrase “panel of potential contributors”, I believe the words “risk factors” would be a better choice of word.
Thank you very much. We corrected it.
There is a need for English editing (moderate). In the pdf file I am attaching, I have highlighted some areas in need of re-writing the sentences or using different words.
Thank you very much. We corrected them.
Comments on the Quality of English Language. The manuscript needs an English edit.
Thank you very much. We corrected it.
Thank you very much

Reviewer 3 Report
Comments and Suggestions for Authors
Dear Editor,
Dear Authors,
I did like to read the present paper submitted to the journal life:
Cholangiocarcinoma-related distant thrombosis: unravelling the dynamic interplay with COVID-19 infection
The paper contains information on thrombosis in a special form of cancer (Cholangiocarcinoma) and the relationship to a CORVID-19 infection. The authors reviewed the literature and analyse all reports on the topic. In fact not spectacular, however the manuscript contains important information. The manuscript is well written and the literature research was comprehensive. The authors balanced their discussion concisely. The overall impression of the manuscript is good, I would recommend publishing the manuscript.
I have only some minor comments:
1. The title of the manuscript could be amended into a more informative phrase.
2. I would recommend changing SARS-COV2 into SARS-CoV-2, which is the most used form.
3. Line 326: change anticoagulant into anticoagulants.
Author Response
Response to Review 3 Comments
Dear Reviewer,
Thank you very much for your time and your effort to review our manuscript.
We are very grateful for providing your valuable feedback on the article.
Here is our response and related amendment that has been made in the manuscript according to your review (marked in yellow colour).
Dear Editor,
Dear Authors,
I did like to read the present paper submitted to the journal life:
Cholangiocarcinoma-related distant thrombosis: unravelling the dynamic interplay with COVID-19 infection.
Thank you very much. We really appreciate it!
The paper contains information on thrombosis in a special form of cancer (Cholangiocarcinoma) and the relationship to a CORVID-19 infection. The authors reviewed the literature and analyse all reports on the topic. In fact not spectacular, however the manuscript contains important information. The manuscript is well written and the literature research was comprehensive. The authors balanced their discussion concisely. The overall impression of the manuscript is good; I would recommend publishing the manuscript.
Thank you very much. We really appreciate it!
I have only some minor comments:
The title of the manuscript could be amended into a more informative phrase.
Thank you very much. We changed the title as follows:
“Superficial vein thrombosis in an asymptomatic case of cholangiocarcinoma with recent history of COVID-19”
I would recommend changing SARS-COV2 into SARS-CoV-2, which is the most used form.
Thank you very much. We corrected it. Thank you
Line 326: change anticoagulant into anticoagulants. Thank you very much. We corrected it. Thank you
Thank you very much.

Round 2
Reviewer 2 Report
Comments and Suggestions for Authors
The authors have successfully addressed my comments. Thank you.